# Study on the Optimization of Territory Spatial “Urban–Agricultural–Ecological” Pattern Based on the Improvement of “Production–Living–Ecological” Function under Carbon Constraint

**DOI:** 10.3390/ijerph19106149

**Published:** 2022-05-18

**Authors:** Ran Yu, Yan Qin, Yuting Xu, Xiaowei Chuai

**Affiliations:** 1School of Economics and Management, Anhui Agricultural University, Hefei 230036, China; sciencehu@163.com; 2Institute of Land and Resources, Anhui Agricultural University, Hefei 230036, China; 3School of Geography and Tourism, Anhui Normal University, Wuhu 241000, China; 18900531575@163.com; 4School of Geography and Ocean Science, Nanjing University, Nanjing 210046, China; chuaixiaowei@163.com

**Keywords:** UAE pattern, PLE function, carbon constraint, NRCA index, FLUS model

## Abstract

The spatial layout of the “Production–Living–Ecological” (PLE) function and the spatial optimization of the “Urban–Agricultural–Ecological” (UAE) pattern are the key points and difficulties in territorial space planning. This paper analyzes their spatial concepts and holds that PLE space is a functional space, while UAE space belongs to a regional space. The optimization of the UAE pattern should be guided by the improvement of the PLE function. Therefore, taking Hefei City, China, as an example, this paper analyzes the evolution of the present UAE pattern, evaluates the PLE function under carbon constraint and then determines the improvement direction of the PLE function and finally simulates the future UAE pattern of territory space. The conclusions are as follows: ① From 2011 to 2019, the urban space increased incrementally, while the agricultural space and ecological space decreased continuously, and the urban space expansion squeezed the agricultural and ecological spaces greatly; ② The PLE functions of four districts in the main city are higher than that of five other counties. After the carbon constraint conditions are included, the PLE functions of the main city were reduced due to the relatively strong capacity of carbon source, while the counties’ increased due to a stronger carbon sink capacity; ③ According to the normalized revealed comparative advantage (NRCA) index, it was determined that the functional improvement direction of each district and county are Yaohai District and Shushan District have comprehensive function as a priority, Luyang District and Baohe District give priority to living–ecological function, Changfeng County, Feidong County, Feixi County and Chaohu County give priority to production–ecological function, and Lujiang County gives priority to ecological function; ④ The simulation results show that 2025 is an important node for the evolution of the UAE pattern. The urban spatial expansion during the “14th Five-Year Plan” period will still bring great pressure on agriculture and ecological spaces, and then, the UAE pattern will continue to be optimized and balanced.

## 1. Introduction

Spatial planning systems have good development history in Europe and America, and the current system is relatively complete. In the world, the concept of “territorial space planning” has not been put forward directly, but various spatial planning systems such as regional planning [1], urban planning [2], land use planning [3], landscape planning [4], habitat management planning [5], etc., are taken as the research objects. However, all kinds of spatial planning are relatively independent and even contradictory, so much of the literature discusses their convergence or integration. For example, Lopes et al. [6] thought that city planning encompasses disciplines related to socio-economic, land use, transport, environment, and others, but these disciplines face communication difficulties and objectives’ divergence due to contradictory interests and isolated evolution. Sangawongse et al. [7] took Wat Ket, Chiang Mai, Thailand, as an example to study its transition of planning mode, from centralized planning to collaborative urban land use planning. Kaczmarek et al. [8] developed a machine learning approach for the integration of spatial development plans based on natural language processing.

A series of guiding documents and standards marked by Several Opinions of the Central Committee of the Communist Party of China and the State Council on Establishing and Supervising the Implementation of the Territorial Space Planning System (issued in 2019, hereinafter referred to as the Opinions) have been issued one after another. Additionally, the preparation of the planning has been carried out in various places all over the country. This shows that the overall framework of China’s territorial space planning system has been basically established. Territorial space planning systems are a major innovation in the field of spatial planning in China, which aims to solve the problems of excessive types, overlapping contents and even conflicts of China’s spatial planning. Different from the strategic design of the international spatial planning system, the Opinions point out that production space, living space and ecological space (Production–Living–Ecological, PLE) should be distributed scientifically, and propose to optimize urban space, agricultural space and ecological space (Urban–Agricultural–Ecological, UAE) from the strategic level.

In China, PLE space was first proposed in 2012 in the 18th National Congress of the Communist Party of China report, while UAE space was first presented in the Pilot Program of Provincial Spatial Planning in 2016. However, there is no clear definition of the two spatial concepts of PLE and UAE, so most of the studies are carried out on PLE and UAE spaces, respectively. The research on PLE space mainly includes functional relationship [9], functional evaluation [10], factor analysis [11] and research on urbanization [12], human settlement [13], land use [14] and ecological environment effects [15] from the perspective of PLE space. For the research of UAE space, mainly based on the spatial demarcation of UAE space, the evolution characteristics [16], scale structure [17] and spatial layout [18] were studied. It can be seen that PLE and UAE spaces have a good research basis, but few studies combine them for research. The main reason could be that PLE and UAE spaces both involve the concept of space, in order to avoid confusion. Through the analysis of the specific research content, it was found that PLE space is more focused on reflecting the function of space. At the same time, UAE space is more consistent with the concept of region and reflected in the land use structure. A production function is represented by regional production, such as agricultural and industrial production, real estate, and trade. A living function is reflected by human and social development—urbanization rate, GDP per capita, average income, and education level. An ecological function mainly refers to the ability of a region to provide ecosystem services and the level of environmental protection. Urban space refers to urban construction and development space—mainly construction land. Agricultural space refers to agricultural production space—mainly cultivated land. Ecological space is a natural landscape space—forest land, grassland, water areas and other land use types. The implementation of the planning is more directly reflected in the regional space.

Therefore, this paper argues that although both PLE and UAE spaces contain the concept of space, their connotations are different. The PLE space is a kind of functional space, which reflects the functionality of products and services that regional space can provide. The scientific distribution of PLE space as stated in the Opinions should focus on improving its functions. The UAE space belongs to the regional space. The optimization of UAE space mentioned in the Opinions reflects the structure and layout optimization of the regional space. In the process of optimization, it should meet the requirements of the regional space to enhance the supply function of products and services. In this sense, PLE space reflects the functionality of UAE space. Additionally, if we avoid the confusion in the expression of the word “space”, we can express it as the optimization of the UAE pattern guided by the improvement of the PLE function. Moreover, with the transformation and development of a region, the dominant and non-dominant functions will also change, so it is necessary to distinguish the current function of the region and its foreseeable functional changes.

Based on this, this study starts from the analysis of the evolution of the present UAE pattern, and on the basis of the evaluation of PLE functions, determines the improvement direction of PLE function through the normalized revealed comparative advantage (NRCA) index, and finally simulates the future UAE pattern of territory space. Among them, the “double carbon” target is considered in the PLE function evaluation, and the carbon constraint index layer is set. In the simulation of the UAE pattern, considering the functional differences among districts and counties, the scenario simulation of districts and counties is adopted. The graphic conceptual framework of the study is shown in Figure 1.

## 2. Materials and Methods

### 2.1. Study Area

Hefei is the capital of Anhui Province, one of the four major science and education cities in China, and a sub-central city in the Yangtze River Delta. It was only a small agricultural town in the early days of the founding of the People’s Republic of China in 1949. The urbanization rate was 30% in 1996, and the GDP just reached CNY 100 billion in 2006. In 2020, the urbanization rate reached 82.28%, and the GDP exceeded CNY one trillion, ranking among the top 20 cities in China. Behind the rapid development of economy and society, a series of pressures were hidden, such as the adjustment of land use structure and industrial structure, the improvement of the carrying capacity of resources and environment, and the optimization of regional main functions. In particular, the current emphasis on the ecological environment and the commitment to the “double carbon” goal posed a great challenge to the new territorial space planning system. Therefore, Hefei is a typical case study, which can serve as a reference for most developing cities. The location map of Hefei is shown in Figure 2.

### 2.2. Data Sources

The data used in this paper include land use data, economic and social statistics data, simulation basic data, carbon accounting data, etc. The land use data were obtained by processing the remote sensing images in 2011, 2015 and 2019 with ENVI5.2, and were divided into six categories according to the land use classification system of Chinese Academy of Sciences: cultivated land, forest land, grassland, water area, construction land and unused land. Social statistics mainly come from statistical yearbooks and statistical bulletins on national economic and social development in all districts and counties. Eight driving factors, such as elevation, slope, aspect, distance to the city center, distance to the town center, distance to the expressway, distance to the main road and distance to the railway, were selected for FLUS simulation basic data, which were derived from OpenStreetMap. Carbon accounting mainly includes six aspects: carbon emissions from energy consumption, carbon emissions from production activities, carbon emissions from waste disposal, carbon emissions from respiration, carbon emissions from food consumption and carbon sinks in terrestrial ecosystems. For specific methods and data processing of carbon accounting, please refer to Yu and Tian’s research results [19].

### 2.3. Modified Constraint Evaluation Model

A traditional multi-factor evaluation model generally adopts a comprehensive weighting algorithm. The basic equation is as follows:(1)$$F=\overset{n}{\underset{i=1}{\sum }}A_{i}\times W_{i}$$
where *F* is the unconstrained evaluation value, *A_i_* is the *i* th evaluation index, and *W_i_* is its weight. Without considering constraints, the algorithm can effectively evaluate the PLE function, but the unexpected output of carbon emissions is difficult to reflect in the evaluation index, which leads to the deviation of the evaluation value of the traditional algorithm from the actual value. Referring to the constraint evaluation model proposed by Hu Xuedong [20], this paper adds constraint *T_c_* to modify the model, and the equations are as follows:(2)$$F^{\prime }=\frac{\sum_{i}^{n}\left[A_{i}\times W{\left(A,T_{c}\right)}_{i}\right]}{T_{c}+1}\;$$
(3)$$T_{c}=\overset{m}{\underset{j=1}{\sum }}C_{j}\times W_{c}\;$$
where *F′* represents the evaluation value under the constraint of carbon emission, *W*(*A,T_c_*) is the weight under the constraint condition, *T_c_* is the carbon constraint index, *C_j_* is the constraint index factor of reacting carbon source/sink, and *W_c_* is the index weight affecting carbon source/sink. In this paper, *W_c_* and *W*(*A,T_c_*) are determined using the entropy method and the analytic hierarchy process, and positive and negative indexes are standardized [21].

### 2.4. Normalized Revealed Comparative Advantage Index

The revealed comparative advantage (RCA) index was put forward by the American economist Balassa, and is used to judge a country’s comparative advantage in international trade [22]. Yu proposed and deduced the normalized revealed comparative advantage (NRCA) index [23], which is more suitable for dynamic, different regional comparison and panel data analysis. It is free from time and space constraints and can realize continuous comparison in time and space. In this paper, it was applied to the discrimination of PLE functional advantages in different districts and counties as the basis of functional improvement direction in subsequent simulation research, and the equation is as follows:(4)$$NRCA_{j}^{i}=\frac{X_{j}^{i}}{X}-\frac{X_{j}X^{i}}{XX}$$
where $X_{j}^{i}$ is the single score of *j* function in *i* area, $X_{j}$ is the total score of *j* function in all areas, $X^{i}$ is the total score of PLE function in *i* area, and *X* is the total score of PLE function in all areas. NRCA > 0 indicates that this function has advantages, while NRCA < 0 indicates that this function does not have advantages.

### 2.5. FLUS Model

The FLUS (Future Land Use Simulation) model is an improvement of the traditional cellular automata model by Liu [24]. It proposes an elaborate self-adaptive inertia and competition mechanism to address the competition and interactions among different land use types. Based on the functional evaluation of the PLE function and referring to the research of Liu [24] and Liang [25], this study simulated the UAE pattern of Hefei using different districts/counties and different scenarios. The model is set as follows:(5)$$TP_{p,k}^{t}=P_{p,k}\times \Omega_{p,k}^{t}\times I_{k}^{t}\times \left(1-sc_{c\to k}\right)\;$$
where $TP_{p,k}^{t}$ denotes the combined probability of grid cell *p* to covert from the original land use type to the target type *k* at iteration time *t*; $P_{p,k}$ denotes the occurrence probability; $\Omega_{p,k}^{t}$ denotes the neighborhood effects; $I_{k}^{t}$ denotes the self-adaptive inertia coefficient; and $sc_{c\to k}$ denotes the conversion cost.

#### 2.5.1. Occurrence Probability

The uniform sampling method was used to sample the land use data and the basic data of driving factors with a sampling resolution of 1 km^2^ × 1 km^2^, and then the basic data of driving factors were normalized. Finally, the probability of the transformation from the original land use type to the target land use type in the evolution of UAE pattern was obtained using the BP-ANN algorithm.

#### 2.5.2. Neighborhood Effects

Neighborhood effects reflected the interaction between different land use types in the UAE pattern and between different land use units in the neighborhood. This study selected the 3 × 3 Moore neighborhood model to calculate:(6)$$\Omega_{p,k}^{t}=\frac{\sum_{3\times 3}con\left(c_{p}^{t-1}=k\right)}{3\times 3-1}\times w_{k}$$
where $\Omega_{p,k}^{t}$ is the neighborhood effect of grid cell *p* at time *t*, $\underset{3\times 3}{\sum }con\left(c_{p}^{t-1}=k\right)$ represents the total number of grid cells occupied by the land use type *k* at the last iteration time *t* − 1 within the 3 × 3 window, and $w_{k}$ is the variable parameter among the different land use types. The range of the parameters is 0–1, and the closer it is to 1, the stronger the expansion ability of the land use type.

#### 2.5.3. Self-Adaptive Inertia Coefficient

A self-adaptive inertia coefficient for each land use type was thus defined to auto-adjust the inheritance of the current land uses on each grid cell according to the differences between the macro demand and the allocated land use amount. The core idea is that if the developing trend of a specific land use type contradicts the macro demand, the inertia coefficient would dynamically increase the inheritance of this land use type to rectify the land use trajectory in the next iteration. The inertia coefficient is defined as:(7)$$I_{k}^{t}=\left\{\begin{array}{c} I_{k}^{t-1}\;\;\;\;\;\;\;\;\;\;\;\;\;\;\;\;\;\;if\;\left|D_{k}^{t-1}\right|\leq \left|D_{k}^{t-2}\right|\\ I_{k}^{t-1}\times \frac{D_{k}^{t-2}}{D_{k}^{t-1}}\;\;\;\;\;\;\;\;\;\;\;\;if\;D_{k}^{t-1}<D_{k}^{t-2}<0\\ I_{k}^{t-1}\times \frac{D_{k}^{t-2}}{D_{k}^{t-1}}\;\;\;\;\;\;\;\;\;\;\;\;if\;0<D_{k}^{t-1}<D_{k}^{t-2} \end{array}\right.$$
where $I_{k}^{t}$ denotes the inertia coefficient for land use type *k* at iteration time *t*, $D_{k}^{t-1}$ denotes the difference between the macro demand and the allocated amount of land use type *k* until iteration time *t* − 1, and $D_{k}^{t-2}$ denotes the difference between the macro demand and the allocated amount of land use type *k* until iteration time *t* − 1.

#### 2.5.4. Conversion Cost

The conversion cost indicates the conversion difficulty from the original land use type to the target type. Based on the result of distinguishing the PLE functional advantages in Hefei, this study set the conversion cost in different scenarios by districts and counties.

## 3. Results

### 3.1. Evolution of Land Use Structure and UAE Pattern

#### 3.1.1. Evolution of Land use Structure

Through the superposition processing of remote sensing images, the evolution results of land use structure in Hefei and all districts and counties were obtained (Table 1 and Figure 3).

From Table 1 and Figure 3, it can be seen that the construction land in Hefei increased by 1.55% from 2011 to 2019, while other land types decreased in different degrees, with the largest decrease in water area, followed by cultivated land, and the decreases in forest land, grassland and unused land were small. From the expansion direction of construction land, it can be seen that it spread to the periphery with the main city and county town as the center. It can be seen that in the rapid urbanization process of Hefei City, the circle expansion mode of urban sprawl was still not completely avoided, which is also the disadvantage of most cities. From 2011 to 2015, in the construction land of the four districts of Hefei’s main city, the area near the main city of Feidong County, Lujiang County and Chaohu County increased significantly, and the supplement of cultivated land was mainly concentrated in Chaohu County and the southeast of Lujiang County. Although the total forest land decreased, the total forest land in western Shushan District and Feixi County increased due to Dashu Mountain National Forest Park and Zipeng Mountain National Forest Park. From 2015 to 2019, the expansion of construction land in and around the main urban area of Hefei slowed down, and it was mainly concentrated in the “Double Centers” of Changfeng County (the northern county and the southern new district). The forest land around Chaohu Lake, the northern part of Yaohai District and Lujiang County, grew significantly, and the supplement of cultivated land was mainly concentrated in the southern part of Feixi County, the eastern part of Feidong County and the northern part of Chaohu County.

#### 3.1.2. Evolution of UAE Pattern

On the basis of land use structure analysis, combined with the meaning of land use classification in the Guide to Classification of Land and Sea Use for Territorial Space Survey, Planning and Use Control (Trial) issued by the Ministry of Natural Resources of the People’s Republic of China in November 2020, the UAE space in Hefei was divided. Among this, the construction land was divided into urban space, cultivated land was divided into agricultural space, and forest land, grassland, water area and unused land were divided into ecological space. The evolution of Hefei’s UAE pattern from 2011 to 2019 is shown in Table 2 and Figure 4.

It can be seen from Table 2 and Figure 4 that the growth in urban space was mainly concentrated in the four districts of the main city and county towns, the growth of agricultural space was mainly concentrated in the southeast of Chaohu County and Lujiang County, the south of Feixi County, the west of Shushan District, the east of Feidong County and the middle of Changfeng County, while the ecological space was mainly concentrated in Feixi County, Lujiang County, the west of Shushan District and the surroundings of Chaohu Lake. In terms of the transferred-out area, the transferred-out area of agricultural space was the largest, with 22,132.91 ha in 2011–2015, which is 2.4 times of the sum of the transferred-out areas of urban and ecological space. From 2015 to 2019, the area of agricultural space transferred out was 17,477.99 ha, which decreased significantly and was basically the same as the sum of that of urban and ecological space. Comparing the transfer-out ratio of the two periods, the urban spatial transfer-out area increased by 1.33%, the agricultural spatial transfer-out area decreased by 0.82% and the ecological spatial transfer-out ratio increased by 1.30%. This reflects Hefei’s achievements in farmland protection and construction land reduced growth, but the city should make more efforts in ecological space conservation.

### 3.2. Evaluation and Advantage Discrimination of PLE Function under Carbon Constraint

In this study, the simulation of UAE patterns aimed to enable the improvement of PLE functions. This subsection evaluates the PLE functions in all districts and counties of Hefei by constructing an index system and determines the direction of function improvement of the districts and counties by advantage discrimination.

#### 3.2.1. Evolution of PLE Function under Carbon Constraint

According to the connotation of the PLE function and referring to the existing research [9,26,27], four primary index layers of production function, living function, ecological function and carbon constraint were constructed, and thirty secondary indexes were screened out (Table 3).

According to the method described in Section 2.3, the overall evaluation results of PLE functions in all districts and counties of Hefei City are shown in Table 4.

It can be seen from Table 4 that the PLE function value of the four districts of the main city was generally higher than that of the other five counties, whether under unconstrained conditions or carbon constraints. After taking into account the carbon constraint conditions, due to the strong carbon source capacity in the main city and the strong carbon sink capacity in the county, the PLE function value in the main city was reduced, while the county level (except Feixi County) was improved, which narrowed the functional value difference between regions. Feixi County, the first county in Anhui Province, ranks among the top 50 counties in China. As mentioned above, Feixi County achieved remarkable results in agricultural space protection and ecological space conservation. Therefore, the PLE functional value of Feixi County experienced a small decrease after carbon constraints were included, which can be used as a reference for the development of environmentally friendly regions. From the point of view of the carbon constraint index, because Yaohai District is a traditional industrial zone, in the development of Binhu New District in Baohe District and the status of Feixi County’s economic and social development, their carbon constraint index showed a fluctuating downward trend. The functional orientation of Chaohu City changed during the “13th Five-Year Plan” period (2016–2020), so its carbon constraint index dropped significantly. Additionally, other districts and counties showed a continuous downward trend. In order to further analyze the functional evolution of PLE functions in each district and county, its single functions were evaluated (Figure 5, Figure 6 and Figure 7).

From the single function value, the production function of Shushan District and Feixi County was obviously higher than that of the other districts and counties, the living function of four districts in the main city was obviously higher than that of the other five counties, and the ecological functions of Shushan District, Yaohai District and Feixi County were obviously higher than those of the other districts and counties. Specifically, for production function, in the four districts of the main city, Shushan was steady and slightly rising, Luyang was steady and slightly declining, Yaohai and Baohe were obviously declining; in the other five counties, Feixi, Changfeng and Feidong were obviously improved and Lujiang and Chaohu basically maintained a wavy stable state. For living function, the four districts of the main city were steady and slightly rising, while the other five counties maintained a wavy stable state. Additionally, the ecological function did not improve; Baohe District, Changfeng County and Chaohu County increased slightly, while other districts and counties basically maintained a wavy stable state. It can be seen that the evaluation results of PLE functions were basically consistent with the previous results of the evolution of UAE patterns, which shows that it can give a good explanation of the evolution of UAE patterns from the perspective of PLE function and can also make the promotion of functional advantages the goal of regional pattern simulations.

#### 3.2.2. Advantage Discrimination of PLE Function

The regional advantage function is not unique, so it was necessary to discriminate the spatial advantage as the basis for the function improvement direction in the subsequent simulation. The NRCA index of each district and county was calculated according to the method described in Section 2.4, and the results are shown in Table 5.

### 3.3. Simulation of Territory Spatial UAE Pattern

Considering the “double carbon” goal and the state’s emphasis on the ecological environment, the districts and counties that do not have the advantage of ecological function were increased with the goal of improving ecological function in the simulation. According to the discrimination results of advantageous functions in Table 5, the scenarios of functional improvement in each district and county were set as follows: Yaohai District and Shushan District give priority to comprehensive function, Luyang District and Baohe District give priority to living–ecological function, Changfeng County, Feidong County, Feixi County and Chaohu County give priority to production–ecological function, and Lujiang County gives priority to ecological function.

#### 3.3.1. Neighborhood Effect and Conversion Cost

In Table 6, the parameter of construction land is set as 1 due to its strongest expansion, and others are set in equal proportion according to the research results regarding Hefei land evolution from 2011 to 2019. In Table 7, according to different functional improvement scenarios, the conversion level of various types of land is set. In principle, a high-grade land use type is not allowed to be converted to a low-grade land use type.

#### 3.3.2. Precision Inspection

The closer the overall accuracy and Kappa coefficient value is to 1, the higher the simulation accuracy is. Additionally, when the Kappa coefficient is greater than 0.8, the model is in a satisfactory state. Based on the present situation of the UAE pattern of Hefei in 2015, according to the discrimination results of PLE function advantages of each district and county in 2019 in Table 5, the UAE pattern in 2019 was simulated, and the simulation results were compared with the present situation in 2019. The overall accuracy and Kappa coefficient of the simulation were calculated to be 0.896 and 0.901, respectively.

#### 3.3.3. Simulation Results

Based on the present situation of the UAE pattern of Hefei in 2019, the pattern evolution simulation was carried out in different scenarios of different districts and counties, and the time was consistent with the current territorial space planning. The simulation results were obtained in three periods: near 2025, medium 2030 and long-term 2035 (Table 8 and Figure 8). For convenience of comparison, the scale and proportion of UAE functions from 2011 to 2035 are shown in Figure 9.

From the simulation results, they are in line with the policies and trends of reduced growth in construction land, cultivated land protection and ecological environment conservation. In terms of scale, the urban space recently increased by 1.9% from 2019 to 2025, continuing the previous incremental growth, but after 2025, the growth trend was obviously gentle and showed a reduced growth trend, with an increase of 0.14% from 2025 to 2030 and 0.05% from 2030 to 2035. The agricultural space remained basically stable, with an average annual decrease of 0.02–0.09%, and the decrease was slightly larger in 2019–2025 compared with several periods. After experiencing a continuous decrease before 2025, the ecological space increased since 2025. In space, urban space is still dominated by urban areas and county towns, ecological space is dominated by Feixi County, Lujiang County and Chaohu County, while agricultural space is still distributed in all counties besides the western part of Shushan District. It can be seen that the “14th Five-Year Plan” period (2021–2025) is still in a phase of rapid economic and social development in Hefei, and the contradiction between development and protection is prominent in land use. During this period, the construction land still needs incremental growth, and the agricultural space and ecological space are obviously squeezed out. Both agricultural space and ecological space are in a shrinking situation, and the reduction rates of both are high year-on-year. The year 2025 is an important time node. After that, the construction land in Hefei will present reduced growth and even show a negative growth trend. The agricultural space was basically stable, and the ecological space shows positive growth. The UAE pattern will continue to be optimized and balanced and enter a period of consolidation and upgrading.

## 4. Discussion

The construction and improvement of the territorial space planning system is the focus of the work of natural resources and related departments at present. The spatial layout of PLE functions and the spatial optimization of UAE patterns are the key points and difficulties. The Opinions only separately put forward vague requirements on PLE functions and UAE patterns but do not specify relationships between them. It is impossible to divide two sets of spatial patterns for a particular area simultaneously in planning practice. Therefore, the research in the existing literature was carried out. The main idea in the study of PLE functions is to take a whole administrative region as the research object, to evaluate the single function of production, living and ecology [10] and to evaluate the coordination of integrated functions based on the single function evaluation [9,11]. However, it is impossible to divide its PLE space for a particular administrative region. Because the concept of the function itself is overlapped, the same area may have these three kinds of functions simultaneously. The performance is the strength of each function—the dominant and non-dominant. Moreover, with transformation and region development, the dominant and non-dominant functions may change [15]. On the other hand, the plan’s implementation is to specifically divide the regional space to guide the development of different regions.

The main idea in the study of UAE patterns is regional division. An administrative region is divided into urban space, agricultural space and ecological space to study the spatial evolution characteristics [16], scenario simulations [28], etc. There is no overlap between the urban, agricultural and ecological spaces, because the division of UAE patterns is based on the premise of land use structure [17]. Therefore, this paper distinguished the spatial concepts of the two and proposed to combine them with functional upgrading as a link. It is considered that the spatial concepts of PLE space and UAE space are different. The former is a functional space, while the latter is a regional space. Simulation and optimization should be aimed at the UAE pattern guided by improving the PLE function.

In the process of research, some ideas for further research were found. First, combined with the research on land use carbon emissions, the restrictions of the “double carbon” target of carbon peak and carbon neutralization on land use type transformation can be added in the simulation process, so that carbon constraints can be simultaneously imposed in the PLE function evaluation and the UAE pattern simulation. Second, it can combine the promotion of PLE function with “Double Evaluation” (the evaluation of resource and environment carrying capacity and the evaluation of suitability for territorial space development, mentioned in the Opinions), and at the same time, it can serve as the guide conditions of UAE pattern simulation. However, there are overlapping parts between PLE function evaluation and “Double Evaluation”, which need to be distinguished by technical means. Third, another research route can be opened by setting goals such as ecological benefit maximization, economic benefit maximization and carbon emission minimization. Especial attention should be paid to the characteristics, differences and realization processes of different goals to guide optimization.

## 5. Conclusions

Taking Hefei as an example, based on the evaluation of PLE function and the discrimination of functional improvement direction of each district and county, this paper simulated the UAE pattern using different districts/counties and different scenarios.

From 2011 to 2019, the urban space in Hefei showed incremental growth, while the agricultural space and ecological space decreased in different degrees. During this period, the expansion of urban space greatly squeezed agricultural space and ecological space, which reflected the pressure on farmland protection, construction land controlling and ecological conservation, which was also the dilemma faced by most developing cities.

The PLE function values in the four districts of the main city were generally higher than those in the other five counties. After the carbon constraint conditions were included, the values of the main city reduced due to the relatively strong capacity of carbon source, while the values of the counties increased due to a stronger carbon sink capacity. Through further calculation of the NRCA index, the functional improvement direction of each district and county was judged. Yaohai District and Shushan District give priority to comprehensive function, Luyang District and Baohe District give priority to life–ecological function, Changfeng County, Feidong County, Feixi County and Chaohu County give priority to production–ecological function, and Lujiang County gives priority to ecological function.

The simulation results show that 2025 is an important node for the evolution of Hefei’s UAE pattern, and the “14th Five-Year Plan” period is still in a period of rapid development and will enter a period of consolidation and upgrading afterwards. From 2019 to 2025, the urban space continued the previous incremental growth state, and then showed a reduced growth situation; the agricultural space continued to be stable, with an average annual decrease of 0.02–0.09%, but the decrease was slightly larger around 2025; and ecological space will continue to decrease before 2025, but it will be increasing from 2025. It can be seen that the expansion of urban space in Hefei during the “14th Five-Year Plan” period will still bring great pressure on agricultural space and ecological space, and then the UAE pattern will be continuously optimized and balanced with the reduction in urban space, the stability of agricultural space and the increase in ecological space.

## Figures and Tables

**Figure 1 ijerph-19-06149-f001:**
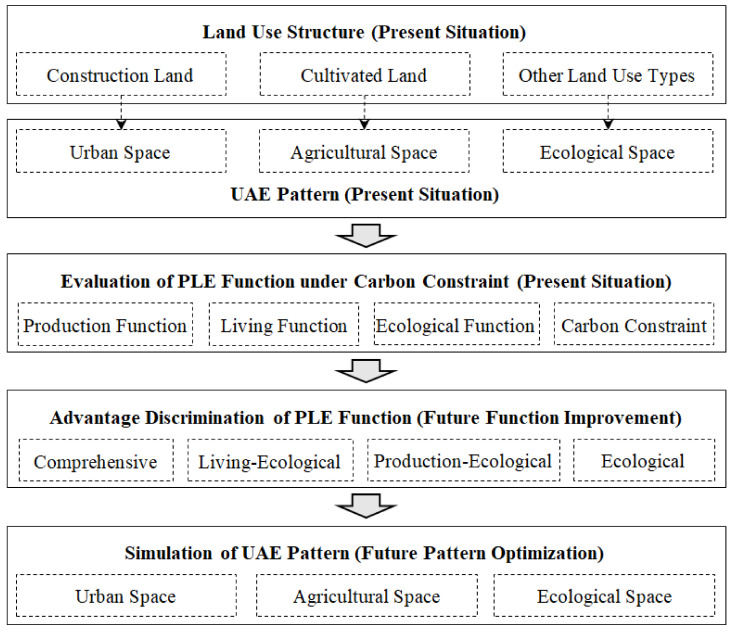
Conceptual framework.

**Figure 2 ijerph-19-06149-f002:**
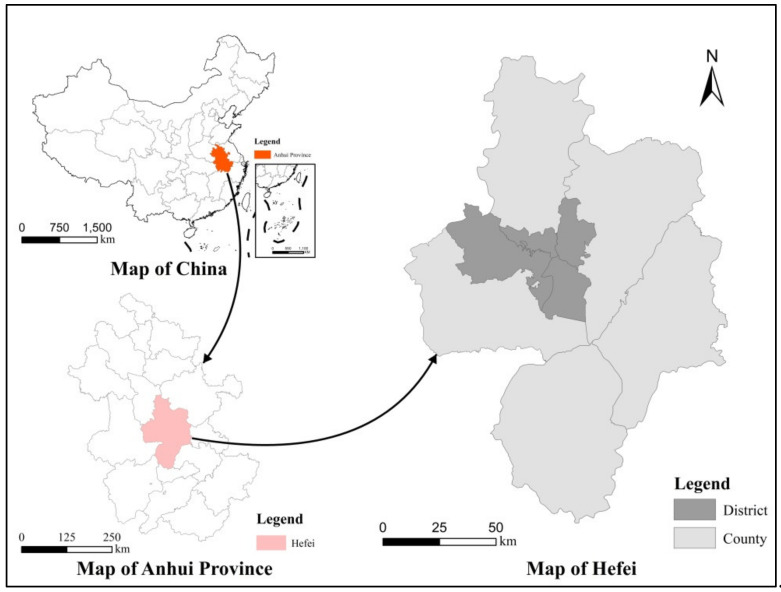
Geographical location of Hefei.

**Figure 3 ijerph-19-06149-f003:**
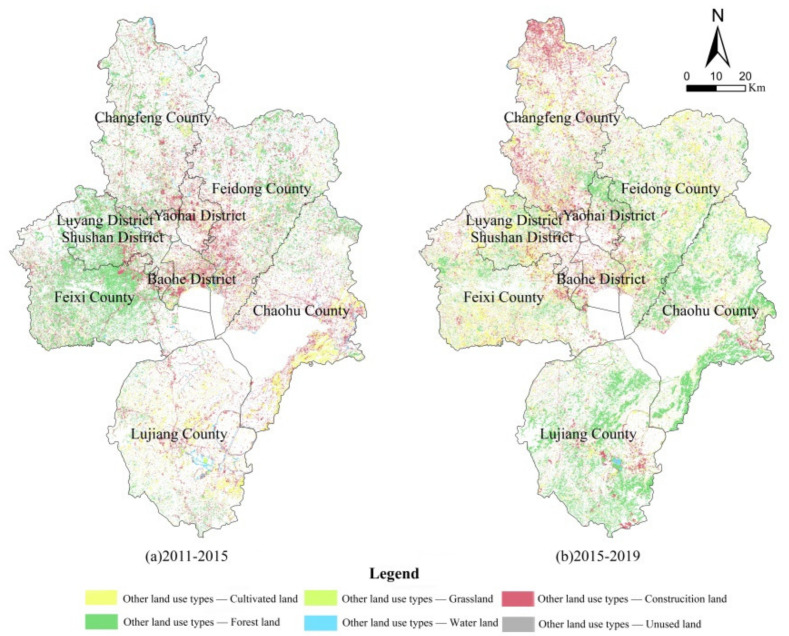
Spatial evolution of land use structure.

**Figure 4 ijerph-19-06149-f004:**
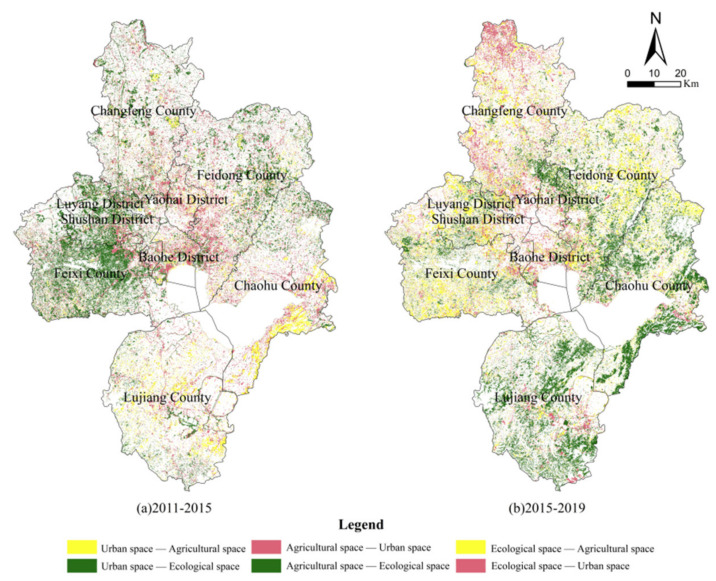
Spatial evolution of UAE pattern.

**Figure 5 ijerph-19-06149-f005:**
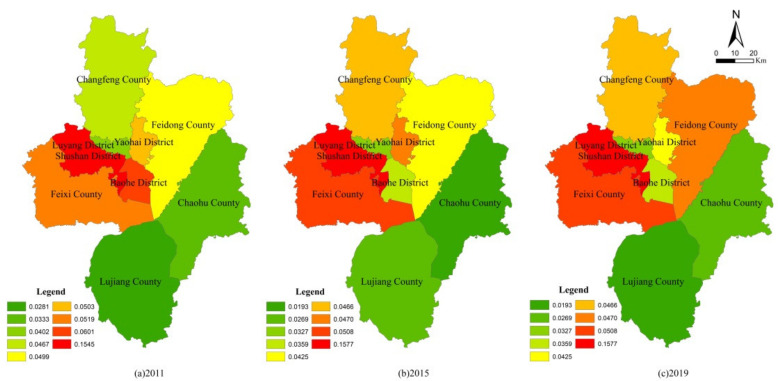
Production function evaluation results of districts and counties under carbon constraint.

**Figure 6 ijerph-19-06149-f006:**
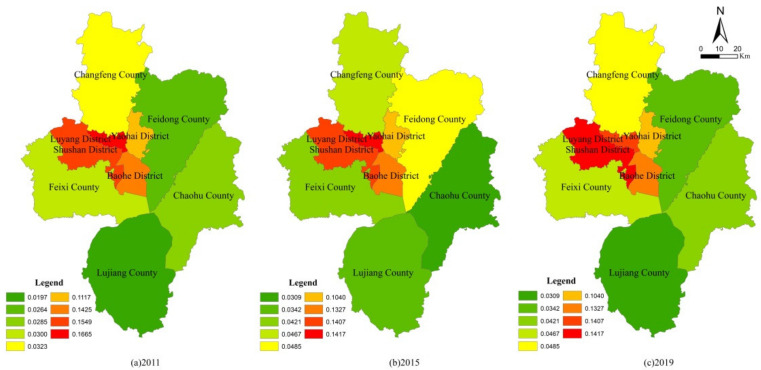
Living function evaluation results of districts and counties under carbon constraint.

**Figure 7 ijerph-19-06149-f007:**
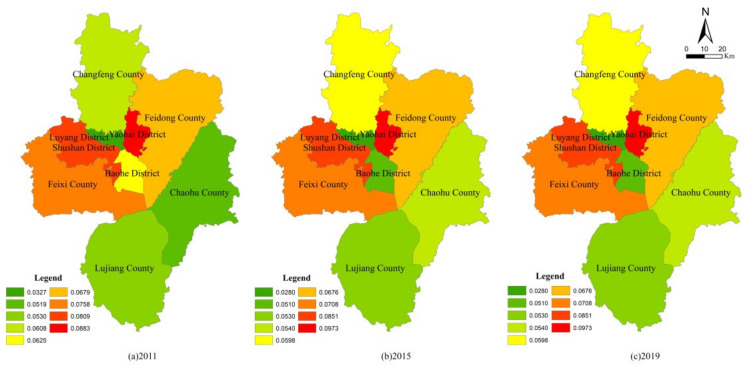
Ecological function evaluation results of districts and counties under carbon constraint.

**Figure 8 ijerph-19-06149-f008:**
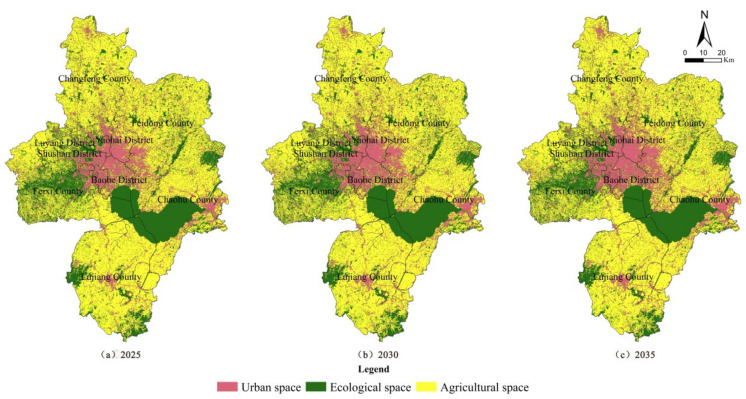
Simulation results of spatial evolution of UAE pattern.

**Figure 9 ijerph-19-06149-f009:**
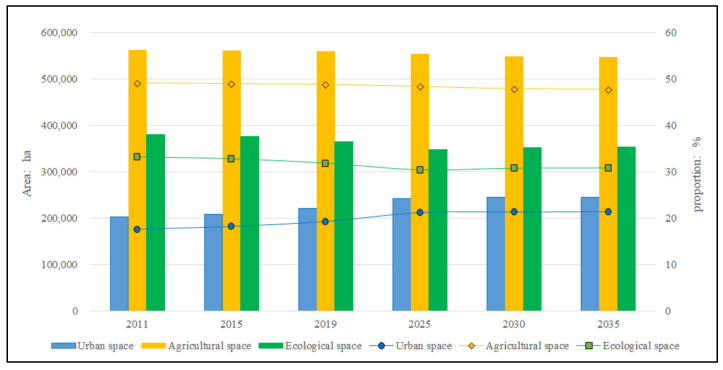
Scale and proportion of UAE from 2011 to 2035.

**Table 1 ijerph-19-06149-t001:** Evolution of scale of land use structure.

Land Use Type	2011		2015		2019		2011–2019	
	Area/ha	Proportion/%	Area/ha	Proportion/%	Area/ha	Proportion/%	Change Value /ha	Amplitude of Variation/%
Cultivated land	561,796.27	49.09	560,457.18	48.97	558,840.00	48.83	−2956.27	−0.26
Forest land	113,574.21	9.92	113,248.66	9.89	112,381.12	9.82	−1193.09	−0.10
Grassland	6855.75	0.60	6460.29	0.56	6344.24	0.55	−511.51	−0.05
Water area	238,074.63	20.80	234,523.50	20.49	225,450.00	19.70	−12,624.63	−1.10
Construction land	203,126.54	17.75	208,636.83	18.23	220,865.00	19.30	17,738.46	1.55
Unused land	21,078.83	1.84	21,179.77	1.85	20,625.87	1.80	−452.96	−0.04

**Table 2 ijerph-19-06149-t002:** Evolution of scale of UAE pattern.

	Transferred from 2011 to 2015/ha				Transferred from 2015 to 2019/ha				Transferred from 2011 to 2019/ha			
	Urban Space	Agricultural Space	Ecological Space	Subtotal	Urban Space	Agricultural Space	Ecological Space	Subtotal	Urban Space	Agricultural Space	Ecological Space	Total
Urban space	0	3176.47	1308.58	4485.05	0	4993.25	2382.77	7376.02	0	8169.72	3691.35	11,861.07
Agricultural space	9328.41	0	12,804.5	22,132.91	5559.01	0	11,918.98	17,477.99	14,887.42	0	24,723.48	39,610.90
Ecological space	1564.29	3236.95	0	4801.24	2766.87	6853.74	0	9620.61	4331.16	10,090.69	0	14,421.85

**Table 3 ijerph-19-06149-t003:** Evaluation index system of PLE function under carbon constraint.

Level I Index	First-Order Weight	Secondary Indexes	Unit	Direction	Secondary Weight	Three-Level Weight	*W*(*A*,*T_c_*)
Production function	0.2667	Gross output value of agriculture, forestry and fishery	CNY one hundred million	+	0.0767	0.0205	0.0181
		Grain yield per unit area	t/ha	+	0.0436	0.0116	0.0150
		Meat output per capita	Kg/person	+	0.0885	0.0236	0.0197
		Industrial added value	CNY one hundred million	+	0.0496	0.0132	0.0208
		Total profits of enterprises above designated size	CNY one hundred million	+	0.1107	0.0295	0.0253
		Floor space of buildings completed	m^2^	+	0.0559	0.0149	0.0170
		Post and telecommunications business income	CNY one hundred million	+	0.0456	0.0122	0.0225
		fixed-asset investment	CNY one hundred million	+	0.0607	0.0162	0.0190
		Economic density	Ten thousand CNY/km^2^	+	0.1141	0.0304	0.0266
		Value of import and export	USD one hundred million	+	0.1760	0.0469	0.0397
		Amount of foreign capital utilized	USD one hundred million	+	0.1786	0.0476	0.0431
living function	0.2667	Population urbanization rate	%	+	0.1629	0.0434	0.0423
		Per capita GDP	CNY/person	+	0.1056	0.0282	0.0255
		Total retail sales of social consumer goods per capita	CNY/person	+	0.1807	0.0482	0.0461
		Average wage of employees on the job	CNY	+	0.1027	0.0274	0.0303
		Per capita savings deposit balance	Ten thousand CNY/person	+	0.2035	0.0543	0.0549
		Education expenditure ratio	%	+	0.0390	0.0104	0.0171
		Per capita residential land area	m^2^/person	+	0.0634	0.0169	0.0197
		Number of beds in health Institutions owned by 10,000 people	Zhang/10,000 people	+	0.1423	0.0380	0.0307
ecological function	0.2667	Per capita park and green areas	m^2^/person	+	0.0943	0.0251	0.0289
		Ground average discharge of industrial wastewater	t/m^2^	-	0.0993	0.0265	0.0255
		Ground average emission of industrial SO_2_	t/m^2^	-	0.1101	0.0294	0.0197
		Ground average amount of chemical fertilizer	t/km^2^	-	0.1548	0.0413	0.0346
		Comprehensive utilization rate of industrial waste	%	-	0.1231	0.0328	0.0541
		Per capita water resources	m^3^/person	-	0.4184	0.1116	0.1038
carbon constraint	0.1999	Per capita carbon emission	tC/a	-	0.1024	0.0205	0.0195
		Ground average intensity of carbon emission	tC/ha	-	0.0800	0.0160	0.0156
		Energy consumption per unit of GDP	kg/CNY ten thousand	-	0.0924	0.0185	0.0211
		Proportion of carbon sinks in terrestrial ecosystems	%	+	0.5744	0.1149	0.1073
		Carbon sink land ratio	%	+	0.1508	0.0302	0.0366

**Table 4 ijerph-19-06149-t004:** Overall evaluation results of PLE function in all districts and counties.

Districts or Counties	2011			2015			2019		
	*T_c_*	*F*	$F^{\prime }$	*T_c_*	*F*	$F^{\prime }$	*T_c_*	*F*	$F^{\prime }$
Yaohai District	0.6574	0.4148	0.3296	0.6615	0.4125	0.3279	0.6440	0.4151	0.3308
Luyang District	0.3824	0.3310	0.2948	0.3611	0.2754	0.2554	0.3444	0.2896	0.2667
Shushan District	0.2759	0.4979	0.4335	0.2709	0.4873	0.4261	0.2704	0.5002	0.4363
Baohe District	0.3840	0.3669	0.3206	0.3591	0.2985	0.2725	0.3776	0.2887	0.2644
Changfeng County	0.3540	0.1892	0.1920	0.3533	0.2072	0.2053	0.2848	0.1972	0.1978
Feidong County	0.3850	0.1998	0.1998	0.3751	0.2181	0.2132	0.3063	0.1914	0.1934
Feixi County	0.3894	0.2190	0.2137	0.4048	0.2300	0.2214	0.3814	0.2296	0.2214
Lujiang County	0.3845	0.1395	0.1563	0.3506	0.1540	0.1660	0.3502	0.1334	0.1507
Chaohu County	0.6636	0.1892	0.1935	0.7364	0.1809	0.1890	0.4234	0.1730	0.1811

**Table 5 ijerph-19-06149-t005:** Discrimination of PLE functional advantages in all districts and counties.

Districts or Counties	2011			2015			2019			Advantage function
	NRCA*_p_*	NRCA*_l_*	NRCA*_e_*	NRCA*_p_*	NRCA*_l_*	NRCA*_e_*	NRCA*_p_*	NRCA*_l_*	NRCA*_e_*	
Yaohai District	−0.0118	0.0070	0.0048	−0.0104	0.0008	0.0096	−0.0147	0.0082	0.0066	comprehensive
Luyang District	−0.0157	0.0399	−0.0242	−0.0117	0.0333	−0.0215	−0.0162	0.0383	−0.0222	living
Shushan District	0.0238	0.0003	−0.0241	0.0325	−0.0101	−0.0225	0.0279	0.0041	−0.0320	comprehensive
Baohe District	−0.0087	0.0209	−0.0122	−0.0125	0.0241	−0.0116	−0.0122	0.0261	−0.0138	living
Changfeng County	0.0037	−0.0128	0.0090	0.0036	−0.0094	0.0058	0.0039	−0.0127	0.0089	production–ecological
Feidong County	0.0048	−0.0170	0.0122	0.0005	−0.0097	0.0093	0.0056	−0.0204	0.0148	production–ecological
Feixi County	0.0038	−0.0180	0.0142	0.0044	−0.0146	0.0101	0.0086	−0.0199	0.0113	production–ecological
Lujiang County	−0.0004	−0.0112	0.0116	−0.0017	−0.0074	0.0091	−0.0004	−0.0117	0.0121	ecological
Chaohu County	0.0005	−0.0092	0.0087	−0.0046	−0.0069	0.0116	−0.0025	−0.0120	0.0145	production–ecological

Note: NRCA*_p_*, NRCA*_l_* and NRCA*_e_* represent the NRCA index of production, living and ecological function, respectively.

**Table 6 ijerph-19-06149-t006:** Assignment of neighborhood effect parameters.

Land Use Type	Cultivated Land	Forest Land	Grassland	Water Area	Construction Land	Unused Land
*w_k_*	0.12	0.05	0.02	0.50	1	0.02

**Table 7 ijerph-19-06149-t007:** Conversion cost matrixes under different scenarios.

	Comprehensive Function						Living–Ecological Function						Production–Ecological Function						Ecological Function					
	a	b	c	d	e	f	a	b	c	d	e	f	a	b	c	d	e	f	a	b	c	d	e	f
a	1	0	0	0	1	0	1	1	1	1	1	0	1	1	1	1	1	1	1	1	1	1	1	1
b	1	1	0	0	1	0	0	1	1	1	1	0	1	1	1	1	1	1	0	1	0	0	0	0
c	1	1	1	1	1	0	0	1	1	1	1	0	1	1	1	1	1	1	0	1	1	1	0	0
d	1	1	0	1	1	0	0	1	1	1	1	0	1	1	1	1	1	1	0	1	0	1	0	0
e	0	0	0	0	1	0	0	0	0	0	1	0	0	0	0	0	1	0	0	0	0	0	1	0
f	1	1	1	1	1	1	1	1	1	1	1	1	1	1	1	1	1	1	1	1	1	1	1	1

Note: a, b, c, d, e and f represent cultivated land, forest land, grassland, water area, construction land and unused land, respectively; 1 means it can be converted, and 0 means it cannot be converted.

**Table 8 ijerph-19-06149-t008:** Simulation results of scale evolution of UAE patterns.

UAE	Land Use Type	2025		2030		2035	
		Area/ha	Proportion/%	Area/ha	Proportion/%	Area/ha	Proportion/%
Urban space	Construction land	242,691.63	21.20	244,269.96	21.34	244,858.33	21.39
Agricultural space	Cultivated land	553,766.02	48.38	548,050.97	47.89	546,776.11	47.77
Ecological space	Forest land	105,658.18	9.23	107,500.56	9.39	107,977.83	9.43
	Grassland	5154.18	0.45	5328.20	0.47	5267.65	0.46
	Water area	218,895.48	19.13	220,150.83	19.24	220,307.68	19.25
	Unused land	18,340.74	1.60	19,205.71	1.68	19,318.64	1.69

## Data Availability

The data presented in this study are available upon reasonable request from the corresponding author. The data are not publicly available due to consent provided by participants.

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
