# Peer review of "Study on the Optimization of Territory Spatial “Urban–Agricultural–Ecological” Pattern Based on the Improvement of “Production–Living–Ecological” Function under Carbon Constraint"

_ijerph, 2022, doi:10.3390/ijerph19106149_

Round 1

Reviewer 1 Report

  • The paper has a very complex methodology, so I propose developing a graphic conceptual framework that will illustrate the use of specified models in the realisation of specific goals and the study's main goal.
  • It is not quite clear to me what the main goal of this article is - is it to find the optimal structure of territory pattern? If so, the conclusions do not show it (in my opinion).

  • In my opinion, the conclusions (including those mentioned in the abstract) focus too much on changes in individual Counties and too little on relationships between different processes/ variables/characteristics. I would instead expect some more general information/conclusions about the observed relationships between the various variables (what influences what? what makes the structure optimal or not optimal ? etc.)

  • In fact, there is no discussion in the article - the discussion should refer to the findings of other authors / other studies. There are only repeated generalities here (in section Discussion).

  • It is also worth paying more attention to a better explanation of the details of the methodology or providing references - e.g. in line 147 the "entropy method" was mentioned without giving any reference to the literature.

Author Response

Thank you very much for your comments and suggestions. Amendments and explanations are as follows:

Point 1: The paper has a very complex methodology, so I propose developing a graphic conceptual framework that will illustrate the use of specified models in the realisation of specific goals and the study's main goal.

Response 1: We draw a conceptual framework (Figure 1) of the paper to illustrate the use of different models.

Point 2: It is not quite clear to me what the main goal of this article is - is it to find the optimal structure of territory pattern? If so, the conclusions do not show it (in my opinion).

Response 2: As described in the last paragraph of the introduction, the purpose of this paper is to optimize the future spatial pattern (UAE pattern) with the guidance of spatial function (PLE function) improving. The conceptual framework (Figure 1) can also express it more intuitively. And the second conclusion of the paper is about function evaluation and function improvement, the third conclusion is about pattern optimization simulation. For pattern and structure, UAE pattern is divided on the basis of land use structure (it described in “Evolution of UAE pattern” in subsection 3.1.).

Point 3: In my opinion, the conclusions (including those mentioned in the abstract) focus too much on changes in individual Counties and too little on relationships between different processes/ variables/characteristics. I would instead expect some more general information/conclusions about the observed relationships between the various variables (what influences what? what makes the structure optimal or not optimal ? etc.)

Response 3: This paper mainly focuses on the overall function of the space and then guides the subsequent pattern simulation. It does not explicitly discuss each function module and its indicators. However, guiding structure optimization by setting various goals, such as maximizing ecological benefits, maximizing economic benefits, and minimizing carbon emissions, could be a direction for further research. It has been added to the discussion section.

Point 4: In fact, there is no discussion in the article - the discussion should refer to the findings of other authors / other studies. There are only repeated generalities here (in section Discussion).

Response 4: The discussion section has been revised.

Point 5: It is also worth paying more attention to a better explanation of the details of the methodology or providing references - e.g. in line 147 the "entropy method" was mentioned without giving any reference to the literature.

Response 5: The entropy method and analytic hierarchy process are widely used, so no literature was added before. A reference has been added, which has a very detailed application of the AHP-Entropy method and includes the standardized treatment of positive and negative indicators.

In addition, some other changes have been made. All changes are marked up using the “Track Changes” function.

Reviewer 2 Report

The paper is interesting, the case study is well presented and the methodology proposed is detailly described. 

Introduction, discussion and conclusion can be improved. I explain better below. 

For a foreign reader is extremely unclear what are the "Urban-Agricultural-Ecological" concept and the "Production-Living_Ecological" concept. The authors wrote in line 64 that there is no clear definition, but that there is some research about these topics. The Authors list the researchers but don't explain the meaning of UAE and PLE. "Functional space" and "regional space" are not enough. Are they legislative concepts or administrative ones? Are they a shared land use or land cover classification? Are they used in land planning or in practices? Who is using these concepts for the first time? I suggest improving the introduction in this sense. 

I suggest that authors carefully consider whether to include these terms in the title.

The confirmation that urban space increasingly from 2011 to 2019 is one of the conclusions of the paper. This phenomenon is evident also in a lot of parts of the world and numerous other authors have written about this (not cited in the paper). The soil consumption needs to be inserted into an international framework almost in the conclusion, if it is not presented in the introduction related to the carbon contrasting. The carbon constraint should also be better framed or summarized. Discussion and conclusion must be improved describing also some other cases in which similar situations are faced. Also, the forecast that urban areas will increase and the agricultural areas will decrease in the future is a well-known issue, but here are presented only as a result of a model. 

In the conclusion, there is no link to the carbon constrain proposed in the title and in the methodology. How does the carbon constraint affect land use and PLE/UAE?

In the Table 1 is unclear the measure unit "area/hm2".

What is the "14th Five-Year Plan"?  

Author Response

Thank you very much for your comments and suggestions. Amendments and explanations are as follows:

Point 1: For a foreign reader is extremely unclear what are the "Urban-Agricultural-Ecological" concept and the "Production-Living-Ecological" concept. The authors wrote in line 64 that there is no clear definition, but that there is some research about these topics. The Authors list the researchers but don't explain the meaning of UAE and PLE. "Functional space" and "regional space" are not enough. Are they legislative concepts or administrative ones? Are they a shared land use or land cover classification? Are they used in land planning or in practices? Who is using these concepts for the first time? I suggest improving the introduction in this sense.

Response 1: The explanations and the summary of the two concepts have been added to the third paragraph of the introduction. It was clarified that the concept of UAE reflects the regional land use structure. The planning implementation is more reflected in the regional land use structure.

Point 2: I suggest that authors carefully consider whether to include these terms in the title.

Response 2: Although the use of the terms PLE or UAE in the title is not uncommon in academic literature, we agree that their use indeed could be confusing for some foreign readers. Nonetheless, we believe that it would be essential to include them in the title to show the purpose of the study immediately.

Point 3: The confirmation that urban space increasingly from 2011 to 2019 is one of the conclusions of the paper. This phenomenon is evident also in a lot of parts of the world and numerous other authors have written about this (not cited in the paper). The soil consumption needs to be inserted into an international framework almost in the conclusion, if it is not presented in the introduction related to the carbon contrasting. The carbon constraint should also be better framed or summarized. Discussion and conclusion must be improved describing also some other cases in which similar situations are faced. Also, the forecast that urban areas will increase and the agricultural areas will decrease in the future is a well-known issue, but here are presented only as a result of a model. In the conclusion, there is no link to the carbon constrain proposed in the title and in the methodology. How does the carbon constraint affect land use and PLE/UAE?

Response 3: The growth of urban space is indeed evident in developing areas. Yet when it reaches a particular stage, the growth rate slows down and even becomes negative as it happened in such cities as Shenzhen, Shanghai, and Beijing in China. The object of this study, Hefei, is still in a rapid development phase. Its rapid development resulted in some problems in the past, such as urban space spread, excessive growth of carbon emissions, etc. However, we do not want to see such results. Therefore, this paper aims to optimize the UAE pattern and restrain urban space’s growth, with the guidance of evaluating and improving the PLE function. The simulation results demonstrate that the Hefei urban space growth will decrease after 2025, which is a good trend. Due to the international emphasis on carbon emission reduction, carbon constraint factors are introduced into the evaluation of the PLE function, namely, the evaluation of the PLE function under carbon constraint. The evaluation results orient the following simulation of UAE. In addition, a conceptual framework is added (Figure 1), to illustrate research models.

Point 4: In the Table 1 is unclear the measure unit "area/hm2".

Response 4: “hm2” is another symbol for hectare alongside “ha”. All “hm2” in the article has been changed to “ha”.

Point 5: What is the "14th Five-Year Plan"?

Response 5: China publishes a five-year plan for economic and social development every five years, and the “14th Five-Year Plan” period is 2021–2025. It is indicated where it first appears in the text.

In addition, some other changes have been made. All changes are marked up using the “Track Changes” function.

Reviewer 3 Report

  • there is no clearly defined purpose of the work and research hypothesis
  • What are PLE and UAE - please try to come up with a clear definition - to European readers, these planning terms operating in China do not say much, making the article incomprehensible. Can these zones be compared in any way to zones in Western Europe or the USA?

  • Agriculture is also associated with greenhouse gas emissions to which CO2 belongs, the question is to what extent the agricultural areas located in the study area emit greenhouse gases, to what extent the mechanization of agriculture will contribute to the increase in CO2 despite the stable share of agricultural land in the functional structure
  • Legend p.6 is not understandable. It should clearly indicated which functions took place in a given period ( is the colour from the first row is the same in the second - so the function has changed?
  • p. 4 (129) "please refer to my research results" - I see there is more than one author of the article so please correct this sentence
  • please improve maps and legends - they are not good quality

Author Response

Thank you very much for your comments and suggestions. Amendments and explanations are as follows:

Point 1: there is no clearly defined purpose of the work and research hypothesis.

Response 1: We have drawn a conceptual framework for the paper (Figure 1), which may visualize this paper’s research design and purpose. This paper aims to simulate the UAE pattern, with the guidance of evaluating and improving the PLE function under carbon constraint (the last paragraph of the introduction describes it). And there are no research hypotheses used in the study.

Point 2: What are PLE and UAE - please try to come up with a clear definition - to European readers, these planning terms operating in China do not say much, making the article incomprehensible. Can these zones be compared in any way to zones in Western Europe or the USA?

Response 2: The third paragraph of the introduction is supplemented with relevant explanations, including the first proposal of the two concepts, the summary of their meanings, etc.

Point 3: Agriculture is also associated with greenhouse gas emissions to which CO2 belongs, the question is to what extent the agricultural areas located in the study area emit greenhouse gases, to what extent the mechanization of agriculture will contribute to the increase in CO2 despite the stable share of agricultural land in the functional structure.

Response 3: We tried your suggestion when writing the paper, that is, to directly add carbon factor into the indicators of production function, living function and ecological function. But it was difficult to achieve, because there would be overlapping indicators. So, we set up a separate carbon constraint layer to constrain the evaluation results as a whole.

Point 4: Legend p.6 is not understandable. It should clearly indicated which functions took place in a given period (is the colour from the first row is the same in the second - so the function has changed?)

Response 4: In Figure 3, the same color represents the result of converting other land use types into a particular land use type. For example, red means that cultivated land, forest land, grassland, water area and unused land are transformed into construction land. Legend indeed was too much, which is not easy to understand, so it has been simplified.

Point 5: p. 4 (129) "please refer to my research results" - I see there is more than one author of the article so please correct this sentence.

Response 5: It has been corrected.

Point 6: please improve maps and legends - they are not good quality.

Response 6: Figure 3 has been modified.

In addition, some other changes have been made. All changes are marked up using the “Track Changes” function.

Round 2

Reviewer 2 Report

Ok, the revisions reported have been fixed or corrected.
Thanks